# Role of Oxidative Stress in Sensorineural Hearing Loss

**DOI:** 10.3390/ijms25084146

**Published:** 2024-04-09

**Authors:** Masato Teraoka, Naohito Hato, Haruhiko Inufusa, Fukka You

**Affiliations:** 1Department of Otolaryngology-Head and Neck Surgery, Graduate School of Medicine, Ehime University, Toon 791-0295, Ehime, Japan; hato.naohito.mh@ehime-u.ac.jp; 2Division of Anti-Oxidant Research, Life Science Research Center, Gifu University, Yanagito 1-1, Gifu 501-1194, Japan; hinufusa@gmail.com (H.I.); y@antioxidantres.jp (F.Y.)

**Keywords:** oxidative stress, sensorineural hearing loss, antioxidants

## Abstract

Hearing is essential for communication, and its loss can cause a serious disruption to one’s social life. Hearing loss is also recognized as a major risk factor for dementia; therefore, addressing hearing loss is a pressing global issue. Sensorineural hearing loss, the predominant type of hearing loss, is mainly due to damage to the inner ear along with a variety of pathologies including ischemia, noise, trauma, aging, and ototoxic drugs. In addition to genetic factors, oxidative stress has been identified as a common mechanism underlying several cochlear pathologies. The cochlea, which plays a major role in auditory function, requires high-energy metabolism and is, therefore, highly susceptible to oxidative stress, particularly in the mitochondria. Based on these pathological findings, the potential of antioxidants for the treatment of hearing loss has been demonstrated in several animal studies. However, results from human studies are insufficient, and future clinical trials are required. This review discusses the relationship between sensorineural hearing loss and reactive oxidative species (ROS), with particular emphasis on age-related hearing loss, noise-induced hearing loss, and ischemia–reperfusion injury. Based on these mechanisms, the current status and future perspectives of ROS-targeted therapy for sensorineural hearing loss are described.

## 1. Introduction

Hearing, which is integral to communication, is a sensory ability shared by most animals, and its loss significantly impairs social life. There are various causes of hearing loss, including congenital hearing loss, which is mainly due to genetic predisposition, age-related hearing loss (ARHL, which will be restricted to sensorineural hearing loss in the context of this review), noise-induced hearing loss (NIHL), and hearing loss caused by drugs and other chemicals, or by ischemia [1]. Hearing loss is not only a physical and financial burden on one’s social life but also contributes to psychological problems and psychiatric disorders, including cognitive decline and depression [2,3]. The growing recognition of hearing loss as a major risk factor for dementia underscores the urgency of addressing it as a global issue [4,5,6,7]. The World Health Organization (WHO) estimates that by 2050, a staggering 2.5 billion people, primarily those over 60 years old, will be living with some degree of hearing loss [8,9,10].

Hearing loss can be caused by various factors; however, damage to hair cells in the cochlea of the inner ear is irreversible. Aging is the most common cause of hearing loss due to this inner ear damage, and its underlying mechanisms are becoming increasingly clear. In 1956, Harman proposed the free radical theory, which states that the production of reactive oxygen species (ROS) and their subsequent damage to biological components are key factors in the aging process [11]. Free radicals are defined as molecular entities or molecular fragments containing one or more unpaired electrons. ROS include both free-radical and non-radical derivatives of oxygen [12]. Because mitochondria are a major source of intracellular ROS, the link between aging and ROS has been the focus of mitochondrial research [13]. ROS production is implicated in several apoptotic and necrotic cell death pathways in the auditory structures [14] and can cause most types of sensorineural hearing loss (SNHL), including ARHL, hereditary hearing loss, ototoxic drug-induced hearing loss (DIHL), and NIHL. In addition to genetic factors, oxidative stress has been identified as a common mechanism underlying several cochlear pathologies, including noise-induced, ototoxic drug-induced, and age-related cochlear degeneration. Oxidative stress and ROS disruption of the redox state have been implicated in cochlear damage [15,16,17]. The cochlea is one of the most susceptible organs to oxidative stress, owing to the high metabolic demands of hair cells in response to sound stimuli. These findings, together with the elucidation of various mechanisms related to hearing loss and ROS, have led to intensive research on the use of antioxidants in the treatment of inner ear disorders. Notably, animal studies are beginning to demonstrate their efficacy; however, the role of antioxidants in human studies remains controversial, with many uncertain results. This review outlines the auditory system and SNHL, mainly due to inner-ear damage, focusing on the relationship between hearing loss and ROS, particularly in ARHL, NIHL, DIHL, and ischemia–reperfusion injury. Based on these mechanisms, we discuss the current status and future perspectives of ROS-targeted therapies for SNHL.

## 2. Auditory System and Sensorineural Hearing Loss

Sound is the result of air vibrations, and the brain has a highly successful mechanism for detecting it [18]. Air vibrations travel to the outer ear and are transmitted to the tympanic membrane and through the middle ear. Connected by three ossicles to the inner ear, the middle ear amplifies the vibrations and transmits them to the inner ear. The cochlea, a part of the inner ear, contains a spiral row of sensory cells called hair cells. Cochlear function is essential for sound recognition in the brain. In humans, the cochlea is a bony organ that forms a two-and-a-half-turn spiral with the cochlear axis at the center. The basilar membrane and organ of Corti exist between the scala tympani and the cochlear duct, whereas the cochlear duct and scala vestibuli are separated by Reissner’s membrane. The scala tympani and vestibuli are connected by a helicotrema at the apical turn. This area contains perilymph fluid, which is similar in composition to normal extracellular fluid. In contrast, the cochlear duct is filled with endolymphatic fluid rich in K+ [19]. The organ of Corti has a single row of inner hair cells (IHCs) and three rows of outer hair cells (OHCs) [20,21]. In humans, there are about 16,000 hair cells, both inner and outer, in the cochlea [22]. These cells have stereocilia, the lower end of which connects to a protein complex containing an ion channel called the mechanoelectric transducer channel. IHCs are mainly innervated by auditory afferent fibers, whereas OHCs are mainly innervated by inhibitory efferent fibers. When sound enters the ear, the footplate of the stapes vibrates, transmitting the vibrations to the basilar membrane [23]. These vibrations cause the IHCs to depolarize, leading to the release of neurotransmitters and the generation of action potentials [24]. OHCs are motile, contracting upon depolarization and expanding upon hyperpolarization [25]. The vibrations of the basilar membrane, enhanced by the OHCs, amplify this electro-mechanical conversion. These electrical signals are transmitted to the brain via the nerves and are ultimately recognized as sound [18].

Hearing loss occurs when any part of the auditory system is affected. There are two types of hearing loss: conductive hearing loss and SNHL. Although SNHL is the most common type of hearing loss, it is usually not treatable by medical or surgical means once the symptoms have passed without improvement. SNHL arises from a dysfunction in the cochlear or nerve pathways involved in hearing. Genetic mutations are critical causal factors of SNHL; several pathologies have also been implicated, including ischemia, infection, intense noise [26,27,28], trauma, aging, the use of ototoxic drugs [29,30], and autoimmunity. Despite the diverse adverse factors, many common factors, such as the influence of ROS and inflammatory cytokines, are believed to be pivotal in causing hearing loss, mainly by damaging hair cells. Treatment with corticosteroids aims to prevent the progression of the damage [31]. Steroids exhibit various physiological activities, including anti-inflammatory and immunosuppressive effects. However, many patients with SNHL do not recover adequately despite optimal treatment [16], warranting the need for an enhanced understanding of the pathophysiology and the development of new treatments [32].

## 3. Reactive Oxygen Species

ROS comprise highly reactive oxygen molecules involved in oxidation reactions, with the most common examples being superoxide (O_2_^−^), hydrogen peroxide (H_2_O_2_), hydroxyl radicals (•OH), and singlet oxygen. The broader term also includes nitric oxide (NO), nitrogen dioxide, and ozone. Although once considered solely toxic to organisms, ROS have been reported to play important beneficial roles as well. When the equilibrium between the production and elimination of ROS is disrupted, affecting cell physiology, this state is termed oxidative stress [33]. Low ROS levels are required for cell proliferation, differentiation, and survival, whereas moderately increased ROS levels can cause DNA damage and promote mutations [34,35,36]. High ROS levels ultimately cause oxidative stress and lead to cellular damage and death. ROS are chemically reactive species containing unpaired electrons and are highly toxic to cells and intracellular structures. It is estimated that ROS are associated with more than 100 clinical symptoms [37,38]. ROS are produced during several processes, including mitochondrial activity in vivo, the oxidation of chemicals and biomolecules, exposure to environmental pollutants such as electrical and UV radiation, and in response to hypoperfusion and reperfusion followed by ischemia [34,36,39]. These ROS are normally detoxified by a variety of antioxidant enzymatic scavengers, such as superoxide dismutase (SOD), catalase, glutathione S-transferase, and glutathione peroxidase (GPX).

ROS include both endogenous and exogenous species, and the major endogenous sources of physiologically relevant ROS include different cellular organs such as mitochondria, peroxisomes, and endoplasmic reticula. Exogenous sources of ROS include smoking, ozone exposure, hyperoxia, ionizing radiation, and exposure to heavy metal ions [40,41]. Mitochondria, the primary source of ROS, generate them as a metabolic byproduct. Mitochondrial ROS have been reported to regulate postmetabolic feedback, autophagy, and inflammatory responses [42,43,44]. Most oxygen is metabolized in the mitochondria, rendering mitochondrial DNA (mtDNA) susceptible to free radical damage. mtDNA has the disadvantages of high information density and low repair capacity, and ROS inhibit mitochondrial transcription. The inner mitochondrial membrane is rich in unsaturated fatty acids but is easily deformed by ROS. This susceptibility to damage stems from the unsaturated nature of the fatty acids, making them prone to peroxidation by ROS. Consequently, the oxidative damage in mtDNA gradually accumulates, leading to cell degeneration and death, which is considered to underlie the progression of aging [45].

## 4. Role of Mitochondrial Oxidative Stress in Hearing Loss

Mitochondria play an important role in ROS production, and genetic mutations affecting mitochondrial function are associated with hereditary hearing loss. Mitochondrial oxidative stress is the common cause of most types of SNHL, including age-related, genetic, and ototoxic drug- and noise-induced hearing loss [46]. Oxidative stress and free radical generation have been shown to contribute to ARHL in inbred mice [47,48,49]. Moreover, superoxide dismutase 1 (SOD1) has been implicated in ROS processing during the oxidative stress response [50]. SOD1 is widely distributed in inner ear tissues, including the spiral ligaments, stria vascularis, and organs of Corti. Notably, SOD1-knockout mice exhibit early progression of ARHL [51]. Similarly, the senescence-accelerated mouse prone 8 (SAMP8) strain, a model for accelerated aging, shows early deafness. Oxidative stress has been implicated in the molecular mechanisms associated with premature cochlear senescence in SAMP8 mice [52]. In these mice, OHCs, spiral ganglion neurons (SGNs), and stria vascularis are reported to gradually degenerate. Moreover, guinea pigs overexpressing catalase in the cochlea display significant protection of hair cells and hearing thresholds after ototoxic treatment [53,54].

The cochlea is extremely susceptible to oxidative stress owing to the high metabolic demands of hair cells in response to sound stimuli. Normally, under physiological conditions, ROS produced in the mitochondria of hair cells are eliminated by the intrinsic antioxidant effects of hair cells. However, under conditions of excessive ROS levels due to external factors, such as noise or ototoxic drugs, the antioxidant defenses of hair cells are compromised, resulting in permanent cochlear degeneration [55,56]. mtDNA is constantly exposed to DNA-damaging agents, similarly to nuclear DNA [57,58]. Owing to its proximity to the electron transport chain and the lack of protective histones, mtDNA is more susceptible to damage from toxic chemicals compared to nuclear DNA [59]. Mutations in mtDNA accumulate and expand during cell division, causing age-related diseases. Oxidative damage to hair cell mtDNA induces mitochondrial dysregulation and triggers apoptosis [60,61]. The activation of the c-Jun N-terminal kinase/mitogen-activated protein kinase (JNK/MAPK) pathway, an apoptotic signaling pathway, has also been observed in OHCs in response to oxidative stress [62]. In addition to apoptosis, ROS generation leads to inflammation and the production of the pro-inflammatory cytokines interleukin-6 [63] and tumor necrosis factor-alpha [64]. The presence of vasoactive lipid peroxidation products, such as isoprostanes, may also contribute to reduced cochlear blood flow associated with excessive noise [65,66]. Noise-induced ischemia and subsequent reperfusion further potentiate ROS [16] (Figure 1).

## 5. Role of Oxidative Stress in ARHL

ARHL, also known as presbycusis, is characterized by a progressive decline in hearing ability with aging [67,68]. Moreover, self-reported hearing loss can be identified in half of those aged 85 years and older [69]. The incidence of ARHL is expected to increase as the older adult population expands [67,70,71]. Several factors have been suggested to influence the onset and degree of ARHL, including genetic factors; racial differences; a history of noise exposure; smoking; alcohol consumption; and various health complications such as diabetes, cardiovascular disease, sex hormones, arteriosclerosis, and obesity [1]. ARHL is thought to result from the age-related degeneration of the cochlea, with cumulative effects of extrinsic damage (noise and other ototoxic agents) and intrinsic disorders (e.g., systemic diseases) [68]. It has been suggested that a substantial contribution to presbycusis accumulates with low-level damage due to noise and other insults [67]. Schuknecht classified four types of age-related changes in hearing based on human temporal bone pathological specimens and audiograms: (1) sensory presbycusis, involving damage to sensory hair cells; (2) neuronal presbycusis, involving damage to SGNs; (3) metabolic presbycusis, involving damage to stria vascularis and leading to strial atrophy; and (4) mechanical presbycusis, characterized by a thick and stiff basilar membrane and cochlear duct [72]. Most patients with presbycusis present with a mixed pathology, and it has been suggested that the central auditory pathways that contribute to the onset and progression of ARHL may be affected, in addition to peripheral pathology.

It is widely accepted that mitochondria are a major source of ROS and represent a crucial site for ROS-induced oxidative damage and that ROS production increases with age [34,73]. The accumulation of glutathionylated proteins with age is an indicator of protein oxidation resulting from the formation of hydrogen peroxide [74,75,76]. Moreover, increased 4-hydroxynonenal indicates lipid peroxidation resulting mainly from the formation of hydroxyl radicals, whereas the presence of 3-nitrotyrosine suggests a peroxynitrite reaction [77]. In rats, mitochondrial deletions increase with age and are correlated with deafness [78]. Additionally, ARHL is more rapid and severe in SOD-deficient mice, suggesting the importance of these endogenous antioxidants in cochlear hair cell survival [48]. A study exploring oxidative stress in the cochlea of aging male CBA/J mice revealed notable insights, including the accumulation of ROS in different tissues of the aging cochlea. Notably, the timing and extent of these oxidative changes varied across the different tissues, suggesting diverse mechanisms at play. These results indicate that different types of oxidative stress are increased in aging cochlea and that the cellular antioxidant defense system is impaired [47].

Acetyl-l-carnitine and alpha-lipoic acid improve cochlear function by reducing the age-related loss of hearing sensitivity. This effect appears to be related to the ability of mitochondrial metabolites to protect and repair age-related cochlear mtDNA damage by upregulating mitochondrial function and improving the energy production capacity [79]. Lecithin is a polyunsaturated phosphatidylcholine (PPC), a high-energy functional and structural element of all biomembranes. PPCs are important antioxidants that protect cell membranes from ROS-induced damage and play a scavenging role in the activation of enzymes such as SOD and glutathione. mtDNA mutations tend to accumulate more frequently than chromosomal DNA mutations, and the same mechanism has been suggested for ARHL in a mouse model of ARHL [80]. Thus, ROS-induced damage to mtDNA may lead to reduced mitochondrial function in the cochlea and consequent hearing loss [81]. In C57BL/6J mice with the deletion of the mitochondrial pro-apoptotic gene *Bak*, age-related apoptotic cell death in cochlear spiral ganglion neurons and hair cells is reduced, and ARHL is prevented. Thus, the induction of a Bak-dependent mitochondrial apoptosis program in response to oxidative stress is an important mechanism of action of ARHL in C57BL/6J mice [82].

## 6. Role of Oxidative Stress on NIHL

NIHL is the second most common cause of SNHL after ARHL, affecting approximately 5% of the global population [26,83]. NIHL can be unilateral or bilateral, and the hearing loss can be transient or permanent [84]. In mammals, sensory hair cells, once damaged, cannot regenerate. Therefore, the noise-induced degeneration of these hair cells and nerves can lead to permanent hearing loss [85]. Studies suggest that OHCs are the primary target of noise-induced damage, which is exacerbated by the loss of OHCs in basal cochlear lesions [86]. Various mechanisms have been postulated as the main causes of noise-induced inner ear damage, including mechanical damage [87], reduced blood flow and hypoxia [88,89,90], glutamate-induced excitotoxicity [91], and free radical-induced tissue damage [89,92,93,94]. Susceptibility to noise can vary among individuals, owing to a mixture of genetic and environmental factors. The unmodifiable risk factors for hearing loss include age, genetics, sex, and race [28,95,96]. Several modifiable risk factors, including failure to use hearing protection [97], smoking [98], physical inactivity [99], diabetes, and heart disease [100], have been associated with an increased risk of NIHL [28].

Excessive ROS production is a widely accepted mediator of noise-induced damage to the cochlea [101,102,103]. ROS production is noted immediately after noise exposure and persists for 7–10 days thereafter, expanding from the basal to the apical direction of the organ of Corti and increasing the area of necrosis and apoptosis [92,104]. Ohlemiller et al. analyzed hydroxyl (OH) radicals in the cochlea and reported an almost 4-fold increase in OH 1–2 h after noise exposure [101]. Yamane et al. further demonstrated this noise-induced increase in free radicals localized to the stria vascularis [93].

Lipid peroxidation products generated by ROS induce apoptosis, and vasoactive lipid peroxidation products, such as isoprostanes, reduce cochlear blood flow [105,106]. Noise-induced ischemia and subsequent reperfusion further promote ROS production [101]. ROS production in the cochlea can also lead to the release of inflammatory cytokines, causing further damage [63,107,108]. NAD(P)H oxidase (NOX), a membrane-bound protein that transfers electrons to oxygen molecules across the cell membrane, has been implicated in noise-induced cellular stress. It may also contribute to ROS production in NIHL, as a reduction in permanent hearing loss has been reported after the intracochlear administration of NOX inhibitors under noise-induced cellular stress [109,110,111,112]. A similar mechanism involving reactive oxygen species has been reported for DIHL [30,113,114,115].

Animal studies have demonstrated genetic susceptibility to NIHL. One strain of mice (C57BL/6J) with ARHL is more susceptible to noise than other strains [116,117,118]. Several knockout mice, including those for *SOD1* [119], *GPX1* [120], and plasma membrane calcium-ATPase pump isoform 2 [121], have been shown to be more sensitive to noise than their wild-type littermates. A study using knockout mice reported genetic deficits that disrupt various pathways and structures within the cochlea, thereby increasing the noise sensitivity of the inner ear [38].

## 7. Role of Oxidative Stress on DIHL

Ototoxic drug exposure is another major cause of SNHL. While there are several classes of drugs that are ototoxic, the most clinically important ototoxicity-associated drugs are platinum-based anticancer drugs such as cisplatin and carboplatin and aminoglycoside antibiotics, which are known to cause irreversible hearing loss [29]. Both classes of drugs primarily damage hair cells in the organ of Corti by producing ROS via apoptotic pathways [122].

The platinum-based drugs cisplatin, carboplatin, and oxaliplatin are among the most widely used anticancer chemotherapeutics. Despite their potential to treat cancer and prolong survival, platinum-based drugs often cause side effects including hearing loss [123]. Cisplatin toxicity in the inner ear is characterized by progressive, bilateral, irreversible hearing loss, particularly in the high frequencies [124]. Chronic changes due to these disorders are seen in the OHCs, stria vascularis and SGCs of the inner ear [125,126,127]. The mechanism has been reported to involve increased ROS levels in the cochlea and the induction of cell death by apoptosis [128,129]. Kopke et al. investigated the antioxidant defense system of the organ of Corti using an in vitro model in rats [130]. Their findings, and those of others, suggest that cisplatin causes damage to hair cells that is associated with the production of ROS, the depletion of intracellular GSH, and interference with antioxidant enzymes in the cochlea [17,131]. Mitochondrial apoptotic pathways have also been implicated in cisplatin ototoxicity, and the inhibition of cell death may be a potential strategy for treating cisplatin-related ototoxicity [132].

Aminoglycosides are among the most commonly used antibiotics to treat infectious diseases; however, they are associated with serious side effects, including nephrotoxicity and irreversible hearing loss [133]. While nephrotoxic side effects are generally reversible, severe ototoxic damage is often not and may result in permanent hearing loss, vestibular dysfunction, or both [134]. One of the major factors in aminoglycoside-induced cochlear damage is oxidative stress via ROS [135]. Gentamicin is known to reduce the mitochondrial membrane potential of OHCs. This leads to the production of NADPH in the OHCs, which increases ROS and induces apoptosis [136]. Aminoglycosides tend to accumulate in the mitochondria of hair cells, which can lead to a pool of ROS and cause hearing loss [137]. The 1555A>G mitochondrial DNA mutation is known to cause hereditary hearing loss associated with aminoglycoside hypersusceptibility [138]. In addition to the A1555G mutation, other mitochondrial DNA mutations associated with hearing loss due to aminoglycoside hypersusceptivity have recently been reported, but the details of their mechanisms of action remain unclear [139].

## 8. Mechanisms of Ischemia–Reperfusion Injury

ROS play an important role in ischemia–reperfusion injury; they are produced during ischemia–reperfusion, inducing organ damage [140,141]. ROS production is intricately orchestrated through the following various mechanisms: (i) Xanthine oxidase: Xanthine produces oxidase from hypoxanthine with xanthine, which is produced via the catabolism of ATP during ischemia. (ii) NOX: NOX is normally divided into a membrane-bound subunit and a cytoplasmic subunit and binds during ischemia–reperfusion. (iii) Mitochondria: During ischemia, the environment within the mitochondria is altered owing to the failure of the electron-transfer system. During reperfusion, the electron-transfer system is reactivated, resulting in electron leakage and the production of ROS from oxygen. (iv) Endothelial nitric oxide synthase (eNOS): Normally, eNOS binds to tetrahydrobiopterin to synthesize NO. However, during ischemia–reperfusion, 7,8-tetrahydrobiopterin binds to NO synthase, and ROS is produced from oxygen. ROS produced by these mechanisms acts on the mitochondrial permeability transition pore (mPTP). The mPTP is normally closed but can be opened by excess ROS generated during ischemia–reperfusion or by low ATP levels [142]. The opening of the mPTP disrupts the normal movement of molecules within the mitochondrial matrix and between the mitochondria and cytoplasm. This can lead to the mitochondria swelling and collapsing [143], ultimately causing a loss of function. Consequently, energy production in the mitochondria is insufficient, the activity of calcium ATPases in the plasma membrane and endoplasmic reticulum is reduced, and intracellular calcium concentration homeostasis is disrupted [144,145]. The ototoxicity of NO is known to be greatly enhanced by its reaction with other toxic agents, especially superoxide, in ischemia–reperfusion injury to form peroxynitrite [146]. In a study on gerbils, transient ischemia caused a remarkable increase in NO production in perilymph, which might be attributable to the inducible NOS pathway [147]. In the same animal model, the antioxidant molecular hydrogen was effective against hearing loss induced by cochlear ischemia, which is thought to be the main cause of idiopathic SNHL [148].

## 9. Potential of Antioxidants for the Treatment of Sensorineural Hearing Loss

Antioxidants have the potential to both preserve and restore hearing function by mitigating mtDNA mutations, as demonstrated in experiments using C57BL/6 mice, a common model for ARHL. Examples of such antioxidants include vitamin C, vitamin E, and melatonin [38,82,149]. Antioxidants are broadly classified as endogenous, produced in vivo, or exogenous, supplied externally. Bipolar antioxidants, such as alpha-lipoic acid, act as antioxidants and restore the antioxidant effects of glutathione, vitamin A, vitamin C, and vitamin E [150]. The administration of alpha-lipoic acid prevents NIHL and carboplatin-induced hearing loss in animals [151,152,153]. N-acetyl-L-cysteine (NAC), a precursor of glutathione, is an endogenous antioxidant enzyme with antioxidant properties [154] that has been reported to reduce the ototoxic effects of noise exposure in animal models [155,156,157,158,159].

Water-soluble antioxidants include methionine, vitamin C, carnitine, riboflavin, niacin, folic acid, polyphenols, and catechins. β-carotene, vitamin E, astaxanthin, and coenzyme Q10 (CoQ10) are widely known as fat-soluble antioxidants and are used as dietary supplements [150]. D-methionine reduces noise-induced oxidative stress and cochlear dysfunction in mice [160]. Folic acid supplementation reduces hearing loss by reducing oxidative stress and homocysteine levels [161]. Vitamin E supplementation reduces cochlear damage in NIHL [162,163], cisplatin [164,165], and gentamicin [166,167]. CoQ10 [168], synthetic analogs of CoQ10, idebenone [169], and soluble CoQ10 are effective in reducing hypoxia-induced hearing loss and NIHL [170]. Studies in guinea pig models of NIHL have shown that combination therapy with magnesium and antioxidants, such as vitamins A, C, and E, may have a protective effect, suggesting potential synergy [171]. Notably, calorie restriction remains the only reliable method for slowing aging in mammals, with numerous reports demonstrating its effectiveness in suppressing age-related diseases and extending lifespan [172,173]. Someya et al. reported that SIRT3, a member of the mammalian sirtuin family localized in the mitochondria, is essential for the suppression of ARHL in mice by calorie restriction [174]. These results suggest that mitochondria-localized mammalian sirtuins play an important role in the suppression of age-related cochlear cell death and ARHL induced by calorie restriction.

As most evidence for the benefits of antioxidants against hearing loss is based on animal studies, their role in humans remains unclear. Prospective studies have not indicated that dietary supplementation with vitamins A, C, or E slows ARHL progression [175,176,177]. Further clinical trials are required to confirm the protective effects of antioxidants against different types of SNHL (Table 1).

## 10. Conclusions

In recent decades, research has revealed the relationship between ROS and ARHL, as well as between ROS and sensorineural hearing loss caused by noise, ischemia, or ototoxic drugs. The urgency for new hearing loss therapies is rising, fueled by evidence from several studies. The cochlea, which plays a major role in auditory function, requires constant high-energy metabolism and is, therefore, extremely vulnerable to oxidative stress, particularly in the mitochondria. While the vast body of literature on this subject is too extensive to be covered comprehensively, the key findings and insights are included in this review. The potential of antioxidants in the treatment of hearing loss has been demonstrated in several animal studies, but results from clinical studies are still insufficient. Randomized controlled clinical trials are required to demonstrate the efficacy of antioxidants in the treatment of hearing loss. Additionally, age-related and noise-, drug-, and ischemia-induced hearing loss have a common cause in the form of ROS, which may prove relevant to prevent the disease. We hope that further clarification of the pathology of hearing loss will lead to its enhanced prevention, as well as that of associated dementia.

## Figures and Tables

**Figure 1 ijms-25-04146-f001:**
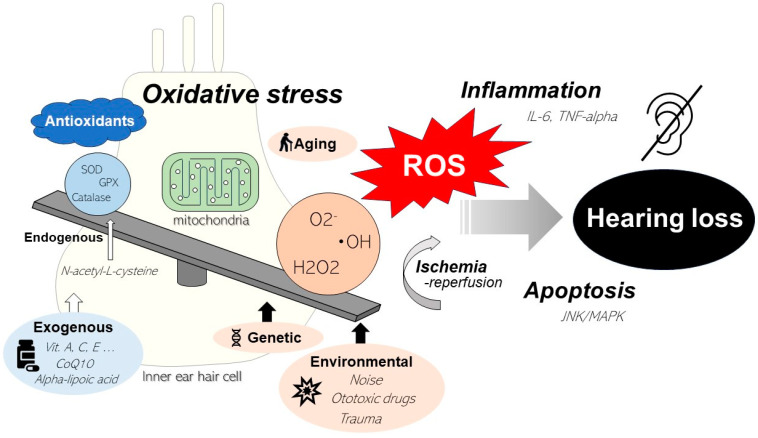
Summary figure of mitochondrial oxidative stress in relation to hearing loss. When the balance between the production and elimination of ROS is disrupted, cellular physiology is affected. Mitochondria play an important role in ROS production. Excessive levels of ROS, caused by external factors such as noise or ototoxic drugs, compromise the antioxidant defenses of hair cells, induce apoptosis, and cause inflammation, resulting in permanent cochlear degeneration.

**Table 1 ijms-25-04146-t001:** Summary of randomized clinical trials on antioxidants for the treatment of hearing loss.

Summary of RCTs of Antioxidants on Hearing Loss in Humans						
Author	Year	Antioxidants	Type of Hearing Loss	Objectives	Sample Size (Patients vs. Control)	Main Findings
Kramer S et al. [178]	2006	N-acetylcysteine	Loud noise	Normal hearing participants	31 (N/A)	No statistically significant differences
L Feldman et al. [179]	2007	N-acetylcysteine	Gentamicin-induced ototoxicity	Hemodialysis patients	40 (20/20)	Significantly more patients exhibiting ototoxicity in the control group
E Kharkheli et al. [180]	2007	Vitamin E	Gentamicin-induced ototoxicity	Acute pulmonary infections	52 (23/29)	No statistically significant differences
Yıldırım M et al. [181]	2010	Salicylate/N-acetylcysteine	Cisplatin-induced ototoxicity	Solid organ tumors	54 (18/18/18)	No difference detected between N-acetylcysteine or salicylate
Lin CY et al. [182]	2010	N-acetylcysteine	Noise-induced temporary threshold shift	Male workers	53 (25/28)	NAC significantly reduced TTS (*p* = 0.03) Effects were more prominent both GSTM1-null and GSTT1-null genotypes.
Tokgoz B et al. [183]	2011	N-acetylcysteine	Ototoxicity drug-induced (Aminoglycosides and vancomycin)	Continuous ambulatory peritoneal dialysis treatment	60 (30/30)	Patients taking NAC had better hearing function test results 4 weeks after the treatment (*p* < 0.05)
Yang CH et al. [184]	2011	Zinc	Idiopathic sudden sensorineural hearing loss	SSNHL patients	66 (33/33)	A significantly larger hearing gain, an increased percentage of recovery, and an increased rate of successful recovery
Hoffer ME et al. [185]	2013	N-acetylcysteine	Blast exposure	Active duty service members	81 (41/40)	In a seven day symptom resolution rate of 86% as compared to 11%
Doosti A et al. [186]	2014	N-Acetylcysteine/Ginseng	Noise-induced	Textile workers	48 (16/16/16)	Reduced noise-induced TTS for NAC and ginseng groups at 4, 6 and 16 kHz (*p* < 0.001)
Kang HS et al. [187]	2014	Vitamin C	Idiopathic sudden sensorineural hearing loss	SSNHL patients	67 (35/32)	HDVC group showed significantly greater complete and partial recovery improvement (*p* = 0.035)
Kopke R et al. [188]	2015	N-acetylcysteine	Military noise during weapons training	Healthy Marine Corps recruit volunteers	566 (277/289)	No significant differences were found for the primary outcome
Villani V et al. [189]	2016	Vitamin E	Cisplatin-induced ototoxicity	Solid malignancies	23 (13/10)	At 1 month, a significant hearing loss at 2k and 8k HZ k was detected in placebo group
Freyer DR et al. [190]	2017	Sodium thiosulfate	Cisplatin-induced	Aged 1–18 years with newly diagnosed cancer	125 (61/64)	The likelihood of hearing loss was significantly lower in the sodium thiosulfate group (*p* = 0.0036)
Kil J et al. [191]	2017	Ebselen	Calibrated sound challenge	Healthy adults aged 18–31 years	83 (22/20/21/20)	Mean TTS at 4 kHz was in the 400 mg ebselen group representing a significant reduction of 68% (*p* = 0.0025)
Brock PR et al. [192]	2018	Sodium thiosulfate	Cisplatin-induced ototoxicity	Hepatoblastoma patients	109 (57/52)	48% lower incidence of hearing loss in the cisplatin-sodium thiosulfate group (*p* = 0.002)
Rolland V et al. [193]	2019	Sodium thiosulfate	Cisplatin-induced ototoxicity	Stage III or IV squamous cell carcinoma	13 (N/A)	Not statistically nor clinically significant differences

## Data Availability

Not applicable.

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
