# Peer review of "Role of Oxidative Stress in Sensorineural Hearing Loss"

_ijms, 2024, doi:10.3390/ijms25084146_

Round 1

Reviewer 1 Report

Comments and Suggestions for Authors

This MS discussed the relationship between sensorineural hearing loss and oxidative stress. The authors first reviewed studies that showed the production of reactive oxygen species (ROS) commonly happens during the pathogenesis of sensorineural hearing loss caused by different causative factors, and then discussed studies that showed the hearing loss treatment or auditory protection potential of antioxidants. Readers can benefit from this kind of review article to systematically understand the current status and progress in this research direction. However, the current version still needs a major revision to reach a good quality.

1.     SNHL, ARHL, and NIHL are categories classified based on different standards. SNHL is hearing loss caused by damage to the sensory cells and/or nerve fibers of inner ear. NIHL is hearing loss caused by exposure to loud noise. NIHL is characterized as sensorineural hearing loss, but ARHL also contains other kinds of hearing loss.  ARHL is defined based on the late onset and the progressive characteristics, it can occur not only because of the changes within the inner ear, but also in other positions like middle ear or along the nerve pathways to the brain. To make the description more accurate, it’s better to have a sentence to clarify that ARHL reviewed in this MS is only restricted to age-related sensorineural hearing loss.

2.     The MS was not nicely organized. The relationship between oxidative stress, ROS, and mitochondria should be clearly discussed in a paragraph at the beginning of the article, and then the same thing doesn’t need to be discussed again and again in the later paragraphs.

3.     The MS should have one paragraph to discuss oxidative stress in ototoxic drug-induced hearing loss.

There are also other issues:

1.     Lines 12 – 13, the form of ROS is not the common cause of these factors. Instead, the form of ROS can be a common downstream pathogenesis factor.

2.     ARHL, NIHL, and ischemia also have genetic factors involved.

3.     Line 74, please cite the reference paper of the “16000 hair cells” calculation and indicate the species.

4.     Lines 88 - 89, genetic mutations are critical causal factors of SNHL, should be included.

5.     Line 127, “mitochondrial endomembrane” is a wrong term, it should be inner mitochondrial membrane.

6.     In lines 129- 131, the statement is incorrect; it should be “the oxidative damage in mtDNA” instead of “mtDNA” itself accumulating. And “mitochondrial inactivity declines” is also illogical here.

7.     The paragraph from line 117 to line 131 needs more accurate citations to support the description.

8.     Line 136, reference 13, is a review paper that discussed mitochondria-dependent and mitochondria-independent pathways of apoptosis; it didn’t mention auditory tissue. The authors should add accurate citations to support the statement.

9.     Lines 142 – 144, ROS or oxidative stress of SAMP8 mouse cochlea should be mentioned.

10.  In lines 152 – 154, this sentence is poorly written. 

11.  There is no legend for Figure 1, only a title.

12.  Lines 169 – 171, which method were hearing thresholds here tested? ABR? Pure tone or click? Please also cite the paper with the data indicated here. 

13.  Lines 195 – 196 and lines 220 – 221, the authors summarized that mitochondrial ROS plays a causative role in ARHL in mammals. This statement is incorrect. Overproduction of ROS by mitochondria plays a role in the pathogenesis of ARHL, but the causative factors of ARHL can be different, such as mutations in specific genes, long-term noise exposure, diseases like high blood pressure and diabetes…

14.  Lines 248 – 250, this sentence should be included in part 3. If the relationship between oxidative stress, ROS, and mitochondria could be clearly discussed in part 3 (the ROS part), then the same thing doesn’t need to be discussed again and again in the later paragraphs.

Comments on the Quality of English Language

English needs to be polished. There are some wired expressions in this MS, I guess because of the language issue.

1.     In lines 75  76, the sentence These cells have stereocilia forming ion channels that open and close in response to sound stimuli is incorrect. I suggest the authors describe stereocilia and then describe mechanotransduction channels located at the tips of shorter row stereocilia.

2.     In lines 117 – 121, the sentence is poorly written. Based on the exogenous factors the authors listed, I guess they wanted to list endogenous and exogenous factors leading to ROS generation. If so, biological processes instead of ROS species should be listed for the endogenous factors or sources.

3.     In lines 122 – 123, the authors might want to indicate the capacity of one free radical attack to initiate a chain reaction. However, the sentence was badly rewritten too. In addition, because of the presence of antioxidant defense systems, most free radical-mediated damages in biological systems are not consequences of chain reactions. The authors should include this information in their review. Because the authors described free radicals here, I suggest they add one sentence to explain the relationship between ROS and free radicals.

4.     In lines 123 – 124, “in vivo” is wrongly used; most oxygen is also metabolized in the mitochondria of in vitro cultured cells. 

Reviewer 2 Report

Comments and Suggestions for Authors

In this manuscript “Role of Oxidative Stress in Sensorineural Hearing Loss”, Teraoka et al. discussed the role of oxidative stress in sensorineural hearing loss and reviewed animals works using antioxidants as protectants against hearing loss. Oxidative stress has long been associated with sensorineural hearing loss caused by environmental insults or aging, and how to alleviate oxidative stress to prevent hearing loss is one of the hot topics in auditory research. This manuscript covers the basics of the auditory system and sensorineural hearing loss, as well as works on oxidative stress in age-related hearing loss, noise-induced hearing loss, and ischemia-reperfusion injury. It’s written in a way easy to follow, and all the literatures were properly cited. There are only a few minor issues. In line 76, the sentence “the afferent and efferent nerve endings are attached to the base of each hair cell” is not accurate; actually, efferent neurons innervate inner hair cell afferent fibers, not inner hair cells. The diagram of hair cells in figure 1 seems awkward, and stereocilia hair bundles are “staircase-like” not hairs of uniform length.

Round 2

Reviewer 1 Report

Comments and Suggestions for Authors

The response letter needed to be carefully prepared. The line numbers cited in the response letter don’t match the revised MS, making it very difficult to review. The authors tried to improve the MS based on reviews’ comments. However, something incorrect can still be found in the revised MS. Please find my detailed comments from the attachment.

Comments on the Quality of English Language

Please find a native English speaker or a professional English polishing service to help you improve your MS language. The MS is not easy to read.

Round 3

Reviewer 1 Report

Comments and Suggestions for Authors

Thank the authors for the revised version.